# Molecular Mechanisms of Seasonal Gene Expression in Trees

**DOI:** 10.3390/ijms25031666

**Published:** 2024-01-30

**Authors:** Xian Chu, Minyan Wang, Zhengqi Fan, Jiyuan Li, Hengfu Yin

**Affiliations:** 1State Key Laboratory of Tree Genetics and Breeding, Research Institute of Subtropical Forestry, Chinese Academy of Forestry, Hangzhou 311400, China; chuxian2016@163.com (X.C.); w524270986@163.com (M.W.); fzq_76@126.com (Z.F.); jiyuan_li@126.com (J.L.); 2College of Information Science and Technology, Nanjing Forestry University, Nanjing 210037, China

**Keywords:** dormancy, annual gene expression, circadian clock, dormancy-associated MADS-box, epigenetics, trees

## Abstract

In trees, the annual cycling of active and dormant states in buds is closely regulated by environmental factors, which are of primary significance to their productivity and survival. It has been found that the parallel or convergent evolution of molecular pathways that respond to day length or temperature can lead to the establishment of conserved periodic gene expression patterns. In recent years, it has been shown in many woody plants that change in annual rhythmic patterns of gene expression may underpin the adaptive evolution in forest trees. In this review, we summarize the progress on the molecular mechanisms of seasonal regulation on the processes of shoot growth, bud dormancy, and bud break in response to day length and temperature factors. We focus on seasonal expression patterns of genes involved in dormancy and their associated epigenetic modifications; the seasonal changes in the extent of modifications, such as DNA methylation, histone acetylation, and histone methylation, at dormancy-associated loci have been revealed for their actions on gene regulation. In addition, we provide an outlook on the direction of research on the annual cycle of tree growth under climate change.

## 1. Introduction

Over the course of an annual cycle, global environmental conditions can change dramatically. Temperature and day length are probably the most important factors in environmental change that undergo annual oscillatory changes. Plants have evolved to adapt to these conditions for seasonal changes such as flowering, germination, leaf growth and senescence, and physiological changes [1]. These annual rhythmic cycles may be a forced oscillatory behavior centered on a process of plant regulation of exogenous environmental factors; during the course of evolution, plants have arisen specific molecular oscillators that contribute to the formation of annual cyclic rhythms (Figure 1A). The endogenous molecular oscillators evolved from adaptation to the environment also play a crucial role in the regulation of rhythmic growth activity [2,3].

The annual cycle activity of tree buds is an important avenue for understanding molecular regulation, which requires the synergistic action of plant endogenous oscillators with the regulation of environmental responses. A number of reviews have addressed this issue in terms of molecular regulation, hormones, and dormancy [2,4,5,6,7]. Research in this area is also receiving increasing attention in the context of global climate change [8,9,10]. Here, we aim to provide an overview for the molecular regulation of annual rhythms by analyzing and summarizing endogenous oscillators, environmental signaling pathways, and epigenetics from the perspective of the establishment of annual cycle gene expression. We believe that the establishment of the annual rhythmic pattern of gene expression is a landmark event in the adaptive evolution of woody plants; understanding and dissecting the regulatory mechanism is of fundamental significance for both basic and applied research.

## 2. Seasonal Gene Expression Underlying the Annual Oscillation of Dormancy and Growth

In order to study the signaling processes of the seasonal perception of temperature and day length, the establishment of annual patterns of gene expression are key to identify underlying regulators [11]. Although rhythmic gene expression may be the result of multiple factors, its assessment can provide information on potential regulation. When studying cyclic expression data, it is often necessary to transform the raw gene expression data with the help of some models of cyclic functions because large-scale experiments are statistically difficult to analyze [12]. There are many methods available for estimating the phase, amplitude, and statistical significance of rhythms in time-series data [13,14]. Additionally, comparisons of the methods on the strength and sensitivity have been tested for diverse data sources; although each method may provide a p-value based the correlations of raw data and model, multiple methods are usually needed for the identification of rhythmic genes.

How can we determine whether there are rhythms in gene expression data? We describe the general process of analyzing oscillation patterns to identify possible rhythmic genes (Figure 1B). Typically, statistical analyses of time-series datasets (e.g., gene expression throughout the year) require the collection of multiple years of data; in addition, the appropriate use of biologically replicated data in experimental designs for consistency analyses can be used as a way to identify rhythms. We further ask whether gene expression patterns are influenced by environmental changes. To answer this question, we can investigate the alterations by comparing peak amplitude, acrophase, and correlations in rhythmic expression models [15]. Unlike the intrinsic rhythms of the circadian clock, studying changes in annual rhythms often requires analyzing their relationship to changes in environmental factors in order to understand the effects of adaptive evolution on plant gene expression [16,17].

## 3. Alterations in Circadian Rhythms Are Involved in Regulating Dormancy and Activity of Seasonal Buds

The timing of daily and seasonal processes in plants is regulated by a circadian clock [18]. The primary function of the circadian clock is to respond to changes in the external environment to adjust endogenous oscillators, thereby enabling the regulation of growth and developmental processes [19,20]. Numerous studies have shown that the circadian clock has a significant impact on life processes such as seasonal development, flowering, photosynthetic performance, leaf aging, leaf movement, biomass accumulation, and responses to biotic and abiotic stresses [20,21,22]. Day length is a key environmental cue and an important marker of seasonal change. In trees, circadian clocks are key factors in anticipating seasonal changes and synchronizing their life activities with the environment by regulating changes in gene expression. In temperate regions, when day length drops below a critical value, known as the critical day length (CDL), perennial trees stop growing and enter a dormant period [23]; by reducing expression of the circadian clock genes, the CDL can be shortened, which delays the period of bud burst [24].

The *Populus* genus, a model species for the study of perennial woody plants, has been the subject of a growing number of studies in recent years, which have demonstrated that the circadian clock is closely related to the seasonal growth activities of trees, such as growth cessation, bud set, cold hardiness, and bud burst [25]. Different from the core components of the circadian clock in *Arabidopsis*, *LHY* (*LATE ELONGATED HYPOCOTYL*) and *CCA1* (*CIRCADIAN CLOCK ASSOCIATED1*), *Populus tremula* × *Populus tremuloides* (*Ptt*) has two *LHY* genes that function similarly to *LHY*/*CCA1* and may have overlapping roles. Additionally, *PttTOC1* (*TIMING OF CAB EXPRESSION1*) suppresses the expression of both *PttLHYs* [26]. These circadian clock genes in poplar operate together to regulate seasonal growth, wood formation, and biomass production levels [26]. As winter approaches, the circadian clock genes undergo dynamic stabilization. If the expression of *PttLHY1* and *PttLHY2* is reduced using RNAi (RNA interference), the plant’s cold tolerance decreases and *CBF1* (*C*-*REPEAT BINDING FACTOR1*, a gene specifically implicated in cold hardening) expression is also reduced, implying that *LHYs* play a part in winter dormancy and protection against chilling injury [24]. After dormancy is released, the expression of *LHYs* continues to promote bud emergence, and the down-regulation of expression results in a significant delay in bud emergence [24]. Furthermore, *LHYs* regulate the growth rate of trees by affecting cytokinin biosynthesis, thereby contributing to safe overwintering [27]. The circadian clock component of trees is typically as conserved as that of the model plant, *Arabidopsis thaliana*. It can be reduced to three reciprocal feedback loops: the morning loop, the central loop, and the evening loop (Figure 2). It can be hypothesized that during annual growth, environmental factors (photoperiod, temperature, etc.) can regulate growth and development by subtle or programmed entrainment of the circadian feedback loops in order to establish annual rhythmic expression patterns (Figure 2).

The circadian clock adjusts to changes in the environment, regulating the expression of a number of genes to ensure the successful timing of plant growth arrest, dormancy establishment, and release. For example, the *CO*/*FT* (*CONSTANS*/*FLOWERING LOCUS T*) module is closely associated with seasonal growth cessation in poplar [28]. In *Arabidopsis*, the *CO* and *FT* are necessary for the day length regulation of flowering, inducing flowering as a response to long days [29]. Research has shown that the regulatory mechanism of the *CO*/*FT* module is conserved between *Arabidopsis* and trees [23]. Unlike *Arabidopsis*, the poplar *FT* gene has two functionally diverged homologous genes, of which *FT1* may have a reproductive initiation regulatory function and *FT2* is a nutrient growth regulator that can maintain growth and prevent bud set [30]. In poplar, the circadian clock can cause the rhythmic expression of *CO* genes. The over-expression of poplar *CO* can directly activate the expression of *FT2*. On the other hand, the core circadian clock gene *LHY2* can activate the nocturnal signaling pathway and directly inhibit the expression of *FT2*, thereby regulating tree growth [31]. At the same time, research into the regulation of poplar dormancy has found that under short day conditions, *SVL* (*SHORT VEGETATIVE PHASE*-*LIKE*, *Arabidopsis SVP* (*SHORT VEGETATIVE PHASE*) homologue gene) is a dormancy promoter. It can promote the deposition of callus at the intercellular junctions by activating *CALLOSE SYNTHASE* (*CALS*), which maintains the dormant state of the buds [32,33].

## 4. Annual Temperature Variations Determine the Seasonal Pattern of Tree Growth

Temperature, another important cue for plants to respond to changes in the external environment, plays a critical role in regulating bud development, dormancy, and vernalization in trees [2]. In a number of tree species, including European chestnuts, apples, pears, and other members of the Rosaceae family, temperature is the primary seasonal stimulus for regulating phenology, with low temperatures inducing bud set and dormancy establishment [25,34]. Recent research indicates that dormancy induction based on day length is only successful within an acceptable temperature range [35,36,37]. In poplar, for example, if the buds are exposed to low temperatures before dormancy is completely set, the dormancy-releasing element activates, leading to impaired bud development and dormancy establishment, as well as inhibiting vernalization. Similarly, higher temperatures also harm dormancy establishment and prompt bud flushing during the pre-dormancy phase [38]. Unfavorable temperature conditions, whether excessively hot or cold, can impede the establishment of dormancy and, ultimately, compromise reproductive success.

After entering dormancy, woody plants in both temperate and subtropical regions are subject to a dual temperature regulation, requiring chilling in winter and forcing in spring. These two factors are negatively correlated and affect the timing of spring phenology [39,40]. In the majority of woody species, low temperature is acknowledged as the primary factor restricting the release of dormancy. The accumulation of effective low temperature during dormancy is referred to as chilling accumulation and the efficient low temperature for ending endodormancy differs among plants [39,41]. In brief, plants can only successfully pass through endodormancy when the quantitative requirement for the chilling requirement has been met. After the cessation of endodormancy, plants require a specific level of accumulated forcing temperatures to initiate bud break and plant expansion, thereby accomplishing growth processes like blossoming and foliage dispersion [42,43,44].

## 5. *DAM* and *SVP*-*like* Genes Are Dormancy Related MADS-Box Genes in Trees

MADS-box is a crucial transcription factor in plants, primarily involved in regulating flowering time, floral organ and seed development, and abiotic stress response [1]. In the *Arabidopsis* MADS-box gene family, *AGL24* and *SVP*, with highly similar sequences, are key genes in the regulation of flower development and anthesis, and are able to influence the flowering pathway by directly regulating the expression of *FT* [45]. However, their functions in *Arabidopsis* flowering are reversed. *svp* mutant plants flower early, overexpressed *SVP* transgenic plants flower late, and *AGL24* is a typical flowering promoter [46,47]. Recent studies on a large number of trees have shown that a number of MADS-box genes are extensively involved in the regulation of bud dormancy and have been classified as the AGL24/SVP subfamily based on phylogenetic and molecular evolutionary analyses. These genes are referred to as *DAM* (*DORMANCY*-*ASSOCIATED MADS*-*BOX*) or *SVP*-*like* [7,48,49].

*DAM* genes were initially identified from the *evergreen* mutants of peach, in which the deletion of six *DAM* genes results in a lack of dormancy [50]. Subsequently, in Rosaceae such as peach, Japanese apricot, and apple, the *DAM* genes have been found to form a subclade close to the *Arabidopsis* AGL24 gene [51,52]. Additionally, tandem duplications were found to form multiple closely related *DAM* genes (Figure 3). In peach and Japanese apricot, six tandem repeats of the *DAM* gene were found [50,53]. In woody plants, *DAM* and *SVP*-*like* genes have been identified in species other than Rosaceae; for example, four *SVP*-*like* genes have been identified in kiwifruit and a gene called *SVL* (*SHORT VEGETATIVE PHASE*-*LIKE*) was identified in hybrid aspen (*Populus tremula L. × P. tremuloides Michx.*) and was related to the regulation of bud break [48,54].

An increasing number of studies on the function of *DAM* and two *SVP*-*like* genes in trees are being conducted to explore the regulatory relationship between such genes and tree dormancy. The expression profiles of many of these genes correlate significantly with the progression of the dormancy cycle, with generally high expression during the establishment and maintenance of dormancy and decreased expression during the bud dormancy release phase [55,56,57,58,59]. Firstly, studies have shown that *DAM* and *SVP*-*like* genes play a role in repressing growth during dormancy and bud break after dormancy release. For example, the expression profiles of the four *SVP*-*like* genes in kiwifruit are associated with dormancy, except for the *AcSVP3* transcript, which does not change significantly throughout the year. Results from heterologous overexpression studies in *Arabidopsis* indicate that these four *SVP*-*like* genes are functionally similar to the *Arabidopsis SVP* genes [54]. They all caused abnormal inflorescence and flower structure in *Arabidopsis*, but only *AcSVP1* and *AcSVP3* were able to delay flowering and complement the loss of function of *AtSVP*. Additionally, none of the four genes can complement the *agl24* mutant, which delays flowering. In addition, the overexpression of *AcSVP2* in kiwifruit inhibits the growth of meristematic tissue and delays bud break [52,60]. Similarly, there are two *SVP*-*like* genes in Japanese apricot, of which *PmuSVP1* delays flowering in *Arabidopsis* [61]. In apple, *MdDAM1* has a significantly cyclical expression profile, the silencing of the gene is required for dormancy release and bud break in spring, and RNA-silenced lines cause them to exhibit a persistent growth phenotype. Moreover, differences in the expression between varieties reinforce its role as a key genetic factor controlling dormancy release in apple buds [62].

Furthermore, *DAM* and *SVP*-*like* genes may repress *FT*-*like* genes. In *Arabidopsis*, the expression of *FT* is able to promote flowering, and it is a key factor in the integration of signals in the flowering pathway. Similarly, in trees, *FT*-*like* genes are functionally conserved and can induce phenotypes such as premature flowering [63,64,65,66]. In these species, *DAM* and *SVP*-*like* genes may regulate bud dormancy by inhibiting the expression of *FT*-*like* genes. For example, the Japanese pear PpDAM1 protein can bind to the promoter region of the *PpFT2* gene, inhibiting its expression. During dormancy, the expression level of *PpFT2* is low, but it increases during dormancy release, which is the opposite of the *PpDAMs* gene [67,68]. Similarly, in hybrid poplar, Myc-SVL binds to the CArG box on the *FT1* promoter, directly inhibiting the mRNA expression of *FT1* [48]. However, more research is required on the regulatory mechanisms between *DAM*/*SVP*-*like* and *FT*-*like* genes.

*DAM* and *SVP*-*like* genes collaborate with plant hormones to regulate tree dormancy. Plant hormones are closely associated with the dormancy process and there is sufficient research to suggest that the establishment, maintenance, and release of dormancy is usually accompanied by dynamic changes in ABA and GA levels, both of which have antagonistic effects [25,69]. The SVL regulation model of hybrid poplar bud dormancy suggests that *SVL* can activate the ABA synthesis rate-limiting enzyme coding gene *NCED3* (*9*-*cis epoxycarotenoid dioxygenase 3*) by transcription to inhibit sprouting. Additionally, the exogenous application of ABA can promote *SVL* expression. *SVL* can also negatively regulate GA by inhibiting GA content and the expression of GA-related biosynthetic genes, ultimately regulating bud dormancy [32,48,70]. Similarly, *PpDAM1* can bind to the *PpNCED3* promoter region, regulating ABA content. ABA can also regulate *PpDAMs* through feedback mechanisms [71,72]. Additionally, in *Pyrus pyrifolia* buds, ABA content regulates the maintenance of pear bud dormancy and the expression of *PpyDAM3* [73]. The expression level of *VvSVP* genes in grapes is high during the dormancy stage and decreases before bud break [74]. This suggests that *VvSVP* genes may play a role in regulating bud dormancy [59]. Poplar that overexpress *VvSVP3* stop growing prematurely under short days, resulting in delayed bud break in early spring. *VvSVP3* regulates the ABA and GA pathways, as well as callus synthesis, to promote dormancy within its network. Furthermore, the exogenous application of ABA has a positive effect on the expression of *VvSVP3* [74]. In summary, many reports have revealed a close correlation between *DAM*/*SVP*-*like* genes and hormones.

Different members of *DAM* and *SVP* were found to form homologous or heterologous complexes. It is expected that a high-order protein complex is formed for the molecular function of DAM-associated functions (Figure 3). In *Arabidopsis*, *SOC1* (*SUPPRESSOR OF OVEREXPRESSION OF CONSTANS1*) has been found to integrate multiple endogenous and exogenous factors to regulate flowering time [75]; in the floral induction process, *SOC1*, together with *AGL24* and *SVP*, is required to regulate floral developmental genes [76,77]. In apricot, the expression profile of the *SOC1* homolog has been found to be highly correlated with bud dormancy, as well as with *DAM* genes; studies of *SOC*-*like* genes in tree peonies, kiwifruit, sweet cherries, and apple have also shown that their expression increases during the period of internal dormancy, suggesting synergistic effects of *SOC1* with *DAM* genes [78,79,80,81]. In addition, SOC1 can form protein complexes with DAM by Yeast Two Hybrid analysis [81,82]. Although the bud dormancy–break cycle is tightly linked to histone modification, it is not clear whether the DAM/SVP complex is able to directly recruit histone modifications.

## 6. Epigenetic Mechanisms of Seasonal Gene Expression

Epigenetic regulation is a crucial mechanism for controlling gene expression in plants. Recent research has demonstrated that modifications of histones, DNA methylation, and non-coding RNAs are extensively involved in regulating the seasonal expression of pivotal genes associated with bud dormancy in woody plants (Table 1) [83,84,85]. H3K4me3 (trimethylation of histone H3 at lysine 4), H4ac (acetylation of H4), and H3ac (acetylation of H3) are recognized as histone marks that are active and likely to promote gene expression, whereas H3K27me3 (trimethylation of histone H3 at lysine 27) and H3K9me3 (trimethylation of histone H3 at lysine 9) are marks that are inactive and often associated with transcriptional repression (Figure 4). Except for H3K9me3, all four types of histone modification have been studied extensively in woody plants (see Table 1 for details). In peach, H3K4me3, H3K27me3, and H3ac are involved in the regulation of *DAMs* genes, which are important for controlling dormancy [86,87,88]. During dormancy release in pear and kiwifruit, a reduced expression of *PpMADS13*-*1* and *AcSVP2* correlated strongly with decreased H3K4me3 levels, but did not significantly change H3K27me3 or H3K9me3 modification levels [60,71]. MADS-box transcription factors are not the only known targets of histone modification during bud dormancy. *PpeS6PDH* (*SORBITOL*-*6*-*PHOSPHATE DEHYDROGENASE*), an important gene for sorbitol synthesis in peach buds, is repressed for expression in dormant flower buds, and this change in expression is accompanied by changes in H3K4me3 and H3K27me3 modifications in specific regulatory regions of the gene [89]. The active histone marker H3K27me3 for *PpEBB1* (*EARLY BUD*-*BREAK1*) in pear was enriched before bud burst, and then the expression level of *PpEBB1* peaked to promote bud break [90]. The H3K27me3 modification *EBB3* in poplar also plays an important regulatory role in the release of the dormant phase [91]. The expression of *AcFLCL* (*FLOWERING LOCUS C*-*LIKE*) in kiwifruit is up-regulated with increased levels of H3K4me3 modification, which promotes bud sprouting [92]. Histone modifications are closely linked to the establishment, maintenance, and release of bud dormancy. In addition to those mentioned above, a wide range of related genes are involved, such as phytohormone biosynthesis and signaling, cell cycle, and cell wall modification pathways [84,93,94].

In addition, many studies have shown that dynamic changes in genomic DNA methylation levels are closely linked to the regulation of the dormancy–growth cycle of the bud. In a number of woody plants, such as chestnut [105], apple [100], poplar [116], almond [101], and sweet cherry [97], the dormant buds often have higher levels of DNA methylation than actively growing buds. DMLs (DEMETER-LIKEs) are a DNA demethylase, and the overexpression of DMLs to reduce methylation levels accelerated dormant bud break in poplar, whereas the down-regulated expression of the transgenic poplar significantly increased DNA methylation levels and delayed bud break [96,116]. The exogenous application of 5-azacytidine (DNA methyltransferase inhibitor) to dormant peony buds was successful in reducing methylation levels, thereby accelerating dormancy release and promoting bud break [117]. Furthermore, the methylation of genes related to ABA or GA (GIBBERELLIN) biosynthesis also affects the regulation of bud dormancy, such as in hybrid poplar, where the repression of GA_4_ biosynthesis involved short-day induced DNA methylation events within the *GA3ox2* (*GIBBERELLIN 3*-*oxidase 2*) promoter [98].

In recent years, there has been a gradual increase in research into the role of non-coding RNAs (ncRNAs) in dormancy. It has been [68] found that microRNA6390 (miRNA6390) may promote the release of endodormancy and bud sprouting by targeting and degrading the transcript of *PpDAM1*, thereby promoting the expression of *PpFT2*. However, in a parallel work, there was no significant difference in the expression of miRNAs after the maintenance and release of endodormancy, nor was the presence of miR6390 detected [118]. This difference in results may be due to the different genetic backgrounds of the varieties. In plants such as grape [113], sweet cherry [111], wintersweet [119], and peony [115], miRNAs were found to be involved in regulating the release of dormancy from dormant buds. Recent studies in peach have revealed that several 21-nt sRNAs (small RNAs) and ncRNAs are induced in *DAM3* and *DAM4* loci, respectively, and this induction was inversely correlated with the down-regulation of homologous *DAMs*. It was also found that the hypermethylation occurring in peach DAMs was closely related to the production of 24-nt sRNA [87]. The regulatory mechanisms involved in sRNAs are often directly linked to the expression of dormancy-associated genes, thus providing an additional level of on/off regulation for dormancy.

## 7. Conclusions and Outlook

The life cycle of trees undergoes periodic events over successive years. Genes with annual rhythmic expression have been found to play an important role in the annual regulation of growth. Acquiring an understanding of this basic information has important implications for our understanding of the regulation of plant growth and development and the evolution of environmental adaptations, and can also be significant for our application to the development of agroforestry.

In recent years, the understanding of the molecular regulatory mechanisms of annual gene expression have gradually been established. However, our information on the regulation of annual rhythmic gene expression is still fragmentary, mainly in: (1) which genes are truly rhythmic and endogenous in their annual phenotypic changes; and (2) our understanding of the role of environmental factors involved in the regulation being far from adequate, such as the superimposed effects of different environmental effects and their corresponding plant response. We believe that the dormancy–activation cycle of buds in forest trees is an important aspect of future research, which involves a number of essential processes such as exogenous environment perception, endogenous hormone regulation, plant signal transduction, meristem activity, organ development, and so on. Future studies, such as in-depth cytological and gene function analyses, environmental adaptation studies, etc., can be instrumental in unraveling the process of molecular regulatory mechanisms.

## Figures and Tables

**Figure 1 ijms-25-01666-f001:**
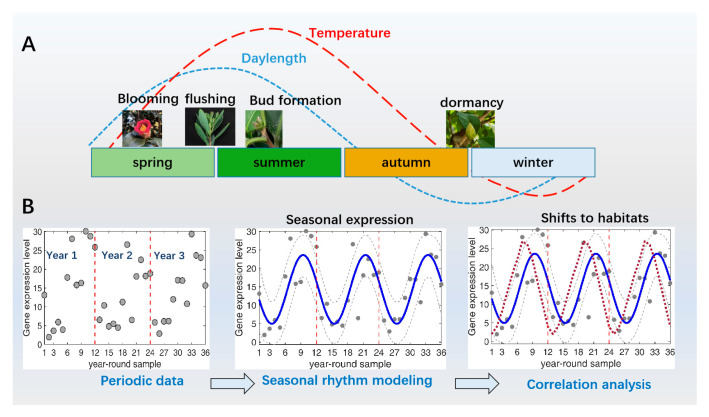
The annual rhythmic cycle of trees. (**A**) Temperature and day length show annual rhythmic variations due to the influence of the Earth’s rotation and revolution. For trees in subtropical and temperate regions, leaf flushing occurs in late spring and early summer. Some of these buds differentiate into flower buds during the summer, which slowly develop and establish throughout the summer and autumn before going dormant in late autumn. When dormancy is released, a suitable spring environment promotes bud burst and flowering. (**B**) General pipeline for identifying and analyzing the gene expression profiles of annual rhythm. The year-round gene expression data of consecutive years are usually noisy and obscure; in order to obtain information about the rhythmicity, some model-based analysis can provide details, including phase, amplitude, and statistical significance. In addition, analyzing the correlation of gene expression in comparison with certain environmental changes can yield information about the shifts of gene expression and seasonality. Blue lines indicate simulated rhythm expression patterns based on model analysis; grey dashed lines indicate confidence intervals; and the red line indicates the expression curve that is altered to adapt to the changing environment.

**Figure 2 ijms-25-01666-f002:**
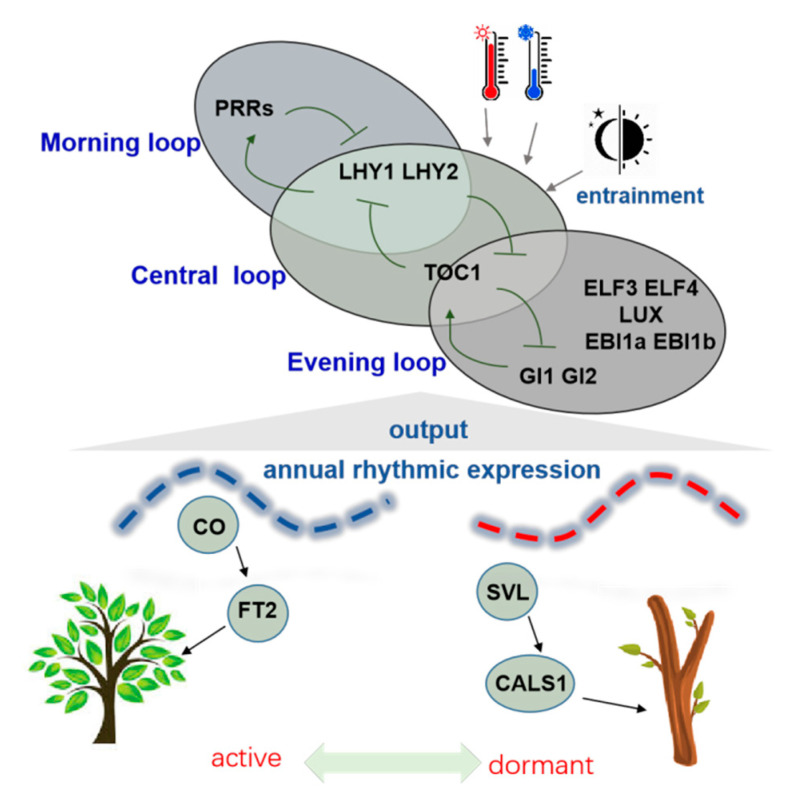
The circadian clock is involved in the establishment of the annual rhythm of gene expression. The simplified system of the circadian clock is presented based on studies of *Populus* and other tree species [25], which is categorized in three major loops, including morning loop, central loop, and evening loop. The core genes of the loops are indicated in the figure: PRRs (PSEUDO-RESPONSE REGULATOR), LHY1/LHY2, TOC1, GI1/2 (GIGANTEA), EBI1a/b (EARLY BIRD), ELF3/4 (EARLY FLOWERING), and LUX (LUX ARRHYTHMO). Temperature and day length are important factors for the entrainment of the parts of the circadian clock, which is thought be critical for establishing annual expression patterns, such as CO and SVL, and, in turn, regulates the entry and exit of bud dormancy.

**Figure 3 ijms-25-01666-f003:**
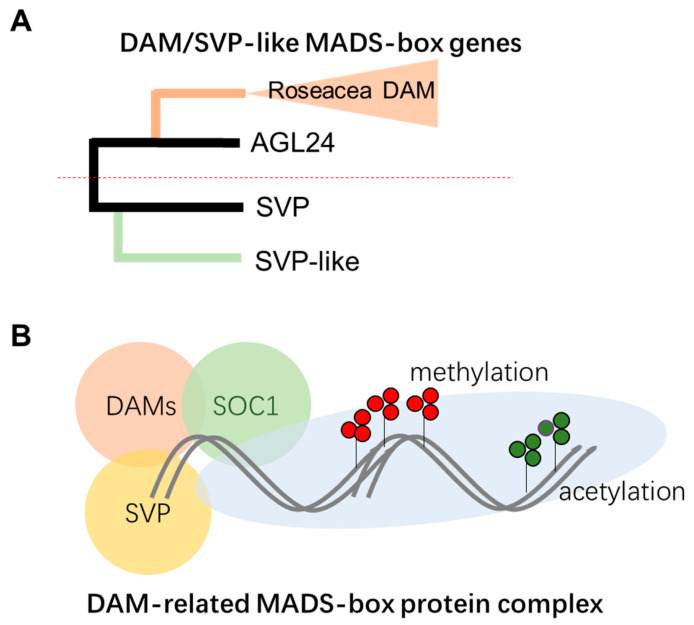
Evolution and function of *DAM*/*SVP*-*like* genes. (**A**) *DAM*/*SVP* genes are a subclade of the MADS-box. A number of analyses in diverse plant lineages suggest that this subclade can be divided into two major groups. The Roseacea *DAM* genes are evolved from the AGL24 group mainly through tandem gene duplication. (**B**) A high-order protein complex is evident for conveying the molecular functions of *DAM*/*SVP* genes. Members of the protein complex contain different types of DAM or SVP- or SOC1-like proteins, and epigenetic modifications of histone are proposed to be involved in the regulation of gene expression through recruiting other factors.

**Figure 4 ijms-25-01666-f004:**
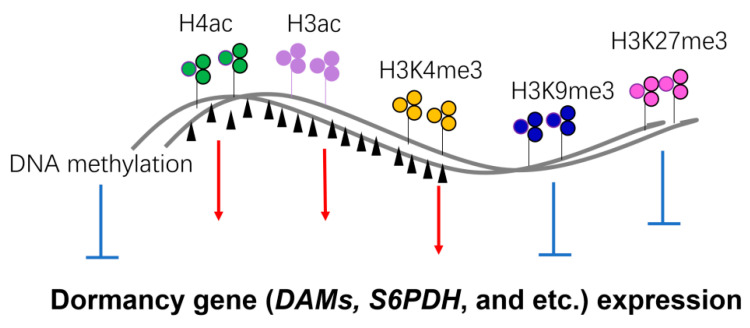
Epigenetic regulation of dormancy-related gene expression. A simple diagram is summarized to include the epigenetic regulation of genes required for dormancy, including DNA methylation (black triangle), H4ac (green), H3ac (purple), H3K4me3 (yellow), H3K9me3 (blue), and H3K27me3 (magenta). The DNA locus of dormancy-related genes (such as *DAMs* and *S6PDH*) is indicated by grey lines. Red arrows indicate the enhancement of gene expression; blue t-shape signs indicate the suppression of gene expression.

**Table 1 ijms-25-01666-t001:** The study of epigenetic modifications in woody plants.

Species	Epigenetic Modification	Mechanism of Gene Expression Regulation	References
*Prunus persica*	DNA methylation	bud early-ripening associates with changes of DNA methylation.	[95]
*Populus*, *Prunus avium*	DNA methylation	global DNA methylation during dormancy–growth cycle.	[96,97]
*Populus*	DNA methylation	DNA methylation represses *GA3ox2* expression.	[98]
*Citrus*	DNA methylation	DNA methylation represses *CcMADS19*.	[99]
*Malus domestica*, *Prunus dulcis*, *Paeonia sufruticosa*	DNA methylation	DNA methylation is linked to chilling responses.	[100,101,102,103]
*Prunus avium*	DNA methylation	DNA methylation and siRNA are involved in the repression of *PavMADS1*.	[104]
*Castanea sativa*	Histone modifications	H4ac is involved in bud set and bud burst.	[105]
*Prunus persica*	Histone modifications	H3K4me3 and H3ac is related with DAM6 activation.	[86]
*Prunus persica*	Histone modifications	The dormancy release induces down-regulation of *PpeS6PDH*, which is associated with a low level of H3K4me3 and a high H3K27me3 level.	[89]
*Prunus persica*	Histone modifications	H3K27me3 is involved in the repression *DAM1* and *DAM5*.	[87]
*Prunus persica*	Histone modifications	changes of H3K4me3 and H3K27me3 levels are involved in the regulation of dormancy.	[93]
*Prunus persica*	Histone modifications	H3K27me3 represses *PpSVP1*, *PpDAM1*, and *PpNCED1*.	[88]
*Pyrus pyrifolia*	Histone modifications	reduction of H3K4me3 and loss of H2A.Z are associated with endodormancy.	[106]
*Pyrus pyrifolia*	Histone modifications	H3K4me3 enhances the expression of *PpEBB*, *DAM*, *PpyGA2OX1*, *NAC88*, and *CYCJ18*.	[90,94,107]
*Populus*	Histone modifications	H3K27me3 represses *EBB3* expression.	[91]
*Actinidia chinensis*, *Citrus*, *Malus domestica*, *Prunus avium*	Histone modifications	H3K4me3 and H3ac enhances the expression of dormancy-related genes (e.g., *DAMs*).	[60,84,92,99,108]
*Prunus persica*	non-coding RNAs	small RNA accumulation pattern is associated with *DAMs* expression.	[87]
*Prunus persica*, *Pyrus pyrifolia*	non-coding RNAs	global identification of noncoding RNAs related to dormancy regulation.	[108,109]
*Pyrus pyrifolia*	non-coding RNAs	miR6390 targets *PpDAM* and regulates *PpFT2* expression.	[68]
*Populus*	non-coding RNAs	identification of dormancy-related small RNAs.	[110]
*Prunus avium*	non-coding RNAs	miR156/SPL, miR172/AP2, and miR166/ATBH15 are involved in dormancy regulation.	[111]
*Malus domestica*	non-coding RNAs	identifcation miR159a-MYB involved in dormancy regulation.	[112]
*Vitis vinifera*	non-coding RNAs	miRNA156, 167, and 1863 are involved in endodormancy	[113]
*Paeonia sufruticosa*	non-coding RNAs	PsmiR172b represses *PsTOE3* during bud dormancy release.	[114,115]

## Data Availability

All data of this manuscript is available on the associated website.

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
