# Peer review of "Molecular Mechanisms of Seasonal Gene Expression in Trees"

_ijms, 2024, doi:10.3390/ijms25031666_

Round 1
Reviewer 1 Report
Comments and Suggestions for Authors
Overall, the manuscript is very interesting and well-written. I understand, of course, that the literature on this subject is extensive, and summarizing the most important aspects can be challenging. However, I find that some parts are a bit confusing and hard to follow, so they should be rewritten. Below are some suggestions.
Lines 31-33: is not clear. May be more of a forced oscillatory behaviour???
Lines 63-70: It doesn't seem like the text answers the question ‘How can we determine whether there are rhythms in gene expression and how they relate to environmental changes?
Line 64: where is Figure 1b?
Line 69: as?
Figure 1: The images are imperceptible, especially the one related to dormancy, which is completely distorted. The text, particularly 'spring,' is unreadable. The caption needs better explanation in relation to the figure sequence. Overall, this figure is unappealing and unclear."
Lines 89-90: not clear
Line 100 LHY or PttLHY?
Line 102: Three (these) circadian clock 102 genes. The ones mentioned above?
Line 106: what is CBE? It is introduced without any explanation.
Lines 112-113: rewrite.
Lines 120-122: not clear
Lines 119-134: This part is crucial, but it is written somewhat confusingly. I recommend expressing it more clearly.
Figure 2 what are CO, FT2 SVL and CALS1? Figures should be self-explanatory.
Line 140: is it herbaceous?
Line 144: in this paragraph well-known information widely reported in textbooks are given.
Lines 170–172: It should be written more precisely and linearly. As it is currently written, it may not be clear to the reader, especially those unfamiliar with the topic, that the well-known MADS-box gene family regulates crucial developmental processes in plants, and the SHORT VEGETATIVE PHASE (SVP) genes are members of this family.
Line 173: In Arabidopsis, there are two major SVP branches, SVP/AGL22 and AGL24. The SVP protein suppresses the flowering process, while AGL24 acts as a flowering activator.
Line 175: DORMANCY-ASSOCIATED MADS-BOX genes (please specify).
Lines 177-186: The text is confusing. The authors should clearly and linearly describe the following: 1) The molecular mechanisms controlling flowering in response to chilling temperatures have been extensively studied in the annual winter plant Arabidopsis thaliana. 2) Recent studies have attempted to establish a connection between MADS-box genes and dormancy and bud break in fruit tree species. 3) Reference the most important bibliography (e.g., https://doi.org/10.3389/fpls.2020.01003) indicating that these genes exhibit similarities with the Arabidopsis AGL24 and SVP or AGL22 genes, and actions of AGL24-like and SVP-like genes in trees.
Figure 3: SOC1 is presented in this figure and caption even before being referenced in the text. It is only by continuing to read the text on page 6 that the nature of SOC1 becomes clear.
Figure 4: blue t-shape signs are not evident.
Line 199: space before homolog
Line 254. repression
Comments on the Quality of English LanguageThe English form is good, but they suggest rewriting some parts more clearly for the reader
Author Response
Thank you for your comments. Please find our point-to-point reponses in the attached file.

Reviewer 2 Report
Comments and Suggestions for Authors
The paper is a valuable review concerning a subject of interest in Plant Science: genetic and epigenetic mechanisms determining bud dormancy/activity in woody plants. The theme is clearly presented in Introduction. The following chapters show the current knowledge on this subject, involving known examples of genes and epigenetic mechanisms that seem to be involved in these processes, correlated with environmental condition. Examples include poplars, Rosaceae species, chestnut, citruses etc. The conclusion is brief and eloquent. The authors have used an impressive reference list, of more than 70 titles, mostly recent papers on the subject.
The only issue I see is the very reference list. I've numbered 72 titles in the text. Below the text, there is a list of 100 titles or so and again one of 72 titles (of which number 4 is "INVALID CITATION"). That seems to be a minute and common error when copy/pasting from a wider list and can be easily remediated. Nevertheless, the authors should carefully check the reference list and correct such errors. Also, please check out correct formatting of references - I see a doi number written as journal issue number (entry 30) or an article with no pagination (entry 9)
Author Response
Thank you very much for your comments.
In this revised manuscript, we double-checked the references. We apologize for not noticing the error in the literature insertion software in the previously submitted manuscript due to our negligence. We have verified the information according to the requirements of the journal formatting. Thanks again.
Round 2
Reviewer 1 Report
Comments and Suggestions for Authors
The manuscript has been appropriately implemented based on the suggestions
Comments on the Quality of English LanguageThe English form is quite good, only minor editing is required